# Improving Deep Learning Anomaly Diagnostics with a Physics-Based Simulation Model

Teemu Mäkiaho *, Kari T. Koskinen and Jouko Laitinen

Unit of Automation Technology and Mechanical Engineering, Faculty of Engineering and Natural Sciences, Tampere University, Korkeakoulunkatu 7, 33720 Tampere, Finland; kari.koskinen@tuni.fi (K.T.K.); jouko.laitinen@tuni.fi (J.L.)
* Correspondence: teemu.makiaho@tuni.fi

**Abstract:** Deep learning algorithms often struggle to accurately distinguish between healthy and anomalous states due to the scarcity of high-quality data in real-world applications. However, these data can be obtained through a physics-based simulation model. In this research, the model serves a dual purpose: detecting anomalies in industrial processes and replicating the machine's operational behavior with high fidelity in terms of a simulated torque signal. When anomalous behaviors are detected, their patterns are utilized to generate anomalous events, contributing to the enhancement of deep neural network model training. This research proposes a method, named Simulation-Enhanced Anomaly Diagnostics (SEAD), to detect anomalies and further create high-quality data related to the diagnosed faults in the machine's operation. The findings of this study suggest that employing a physics-based simulation model as a synthetic-anomaly signal generator can significantly improve the classification accuracy of identified anomalous states, thereby enhancing the deep learning model's ability to recognize deviating behavior at an earlier stage when more high-quality data of the identified anomaly has been available for the learning process. This research measures the classification capability of a Long Short-Term Memory (LSTM) autoencoder to classify anomalous behavior in different SEAD stages. The validated results clearly demonstrate that simulated data can contribute to the LSTM autoencoder's ability to classify anomalies in a peripheral milling machine. The SEAD method is employed to test its effectiveness in detecting and replicating a failure in the support element of the peripheral milling machine.

**Keywords:** Simulation-Enhanced Anomaly Diagnostics (SEAD); physics-based simulations; diagnostics; anomaly classification; synthetic training data; deep learning; LSTM

## 1. Introduction

In manufacturing systems, identifying potential problems early on is vital for safety and cost-effectiveness. Detecting issues promptly not only ensures a secure operational environment but also helps prevent financial losses by addressing concerns before they escalate [1]. This is particularly critical in high-energy processes, where machine breakages or anomalies in machinery may pose a threat to the overall process operation and to the safety of the operating personnel.

Physics-based simulation models and various machine learning tools are used to detect anomalies by analyzing data and recognizing unusual patterns or behaviors that could signal potential risks for failures. However, the identification success depends heavily on the availability of representative training data [2] together with the inclusion of good-quality data to avoid inaccurate analytics and unreliable decision-making [3]. Poor-quality or inconsistent data can lead to inaccurate results, which can lead to false-negative or false-positive occurrences in the classification process. Without proper data, it is difficult for machine learning algorithms to distinguish between normal and abnormal dispersion, whereas creating understandable representations of the data makes it easier to extract useful information when constructing classifiers or other predictive models [4].

Physics-based simulation models are used across industries to create a baseline for a machine system's normal operation and to observe deviations from normal behavior by simulating the system in a computer environment. By understanding the baseline performance of the machine system, deviations from that baseline can be quickly identified and addressed. Also, physics-based solutions may provide additional information in the absence of real data, whereas the absence of breakdown samples prompts a need for further investigation into the development of effective methods to detect anomalies in production data, rather than solely focusing on predicting fault events [5]. Krzyzanska and Nachman focused on generating background and signal events with their simulation model for the training of machine learning classifiers for distinguishing anomalies from normal operation data [6]. A model-based method for real-time anomaly detection that contained a dynamic regression model and adaptive anomaly threshold was trialed by [7] for the forecast of electricity generation needs, load frequency control, and demand forecast in the power industry. In the aviation industry, Simon and Rinehart [8] presented model-based methods for anomaly detection for use with aircraft engines by comparing the outputs observed from an engine with the outputs predicted by their physics-based simulation model. Their research results indicated that making improvements to the trim-point information based on the simulation-model results also improved the effectiveness of anomaly detection.

Various machine learning (ML) techniques are also used to detect anomalies in machine systems. In industrial applications, the ML model's ability to recall past events is essential because it helps to identify patterns and trends in data sets that can be used to make predictions of future events or outcomes. A well-known artificial neural network (ANN) used in deep learning is called the recurrent neural network (RNN) and is widely in use with time series data [9,10]. RNNs can remember information over long sequences of inputs and can process data one element at a time while retaining information about the previous elements. One application of an RNN is Long Short-Term Memory (LSTM), which can be used in regression as well as in a classification type of problem-solving [11,12]. Supervised ML classification performance can be evaluated by comparing the true positives, false positives, true negatives, and false negatives between the labeled true states and the predicted states. From these results, it is possible to use scoring metrics such as the precision and recall of the system, which indicate how well the system is performing in terms of correctly identifying anomalies [13,14].

This study analyzes in vivo data from an extemporary time interval of a peripheral milling machine's manufacturing operation. In vivo data is defined here as referring to data collected from real-world situations that were not primarily created or modified for the collection of data [2]. This may include data collected from experiments, surveys, or observations of naturally occurring phenomena. The data used in this research consists of normal operation data as well as anomalistic behavior data, yet the exact temporal appearance of the anomaly states and the cause of the recognized quality issues were unknown in the initial situation. The milling spindle of the peripheral milling machine consists of 72 milling blades, which are divided into four blade rows as depicted in Figure 1, with each row containing 18 individual blades per spindle circumference. The spindle rotates in a down-feed direction in reference to the profile-feed direction. Blade Rows 1 and 2 are at a 90-degree angle towards the milled profile whereas rows 0 and 3 are at an 86-degree angle, rendering two surface options for welding.

The ranges of the milled-profile height, thickness, and length are depicted in Table 1.

**Table 1.** Milled-profile dimensions.

| Height (mm) X–Direction | Thickness (mm) Y–Direction | Length (mm) Z–Direction |
|:---:|:---:|:---:|
| 70–200 | 5–30 | 6000–23,800 |

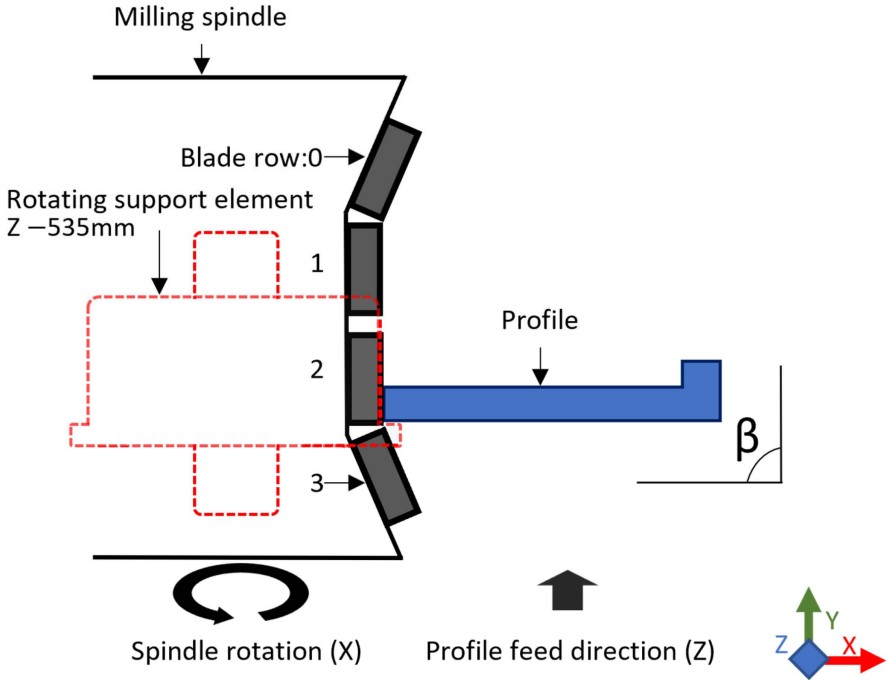

**Figure 1.** Profile angle towards the support element and milling blade under normal operation. The blade rows are assigned numerical designations ranging from 0 to 3.

While in operation, the profile is resting on support elements on both sides of the profile, yet only the spindle side of the support element is annotated in Figure 1. The support element consists of a fixed rod and a bearing rotating the support element, which is in perpendicular contact with the milled profile. The last point of contact between the milled material and the support element occurs in the Z–direction at 535 mm before the profile contact with the milling blades.

Generally, by designing an appropriate rake angle for milling, the amount of material removed with each pass of the milling blade is optimized, which reduces the friction and frictional stress on the workpiece. Improper rake angle increases energy consumption, accelerates tool wear, and accumulates the risk of failure [15]. Akparibo and Normanyo discovered that the amount of electricity consumed is determined by the velocity of the drive and the amount of resistance it meets while in motion [16]. A failure in the profile-support element was discovered in the fault-finding process. The support-element-failure outcome is illustratively visualized in Figure 2. The failure in the support element's bearing induces negative deviation ($\Delta\beta$) to the nominal support angle 'β', causing profile contact angle decrement (Y−) towards the lower blade's Row 3. Thus, due to the milling angle change, the torque demand in the motor will increase because the variable frequency drive (VFD) is aiming to maintain the required cutting velocity. This demand leads to increased electrical-energy consumption in the electrical motor running the spindle as the cutting speed decreases. Consequently, achieving the same amount of work demands a higher energy input.

The first contribution of this study is to report an anomaly detection process for in vivo data that uses a physics-based simulation model to create a baseline for normal behavior and to distinguish any recognized anomalous behavior into binary classes of normal and failure states. The second contribution is to establish an authentic simulated representation of the machine's normal and anomalous torque behaviors. These simulated behaviors are used to improve the classification capability of the LSTM autoencoder when more descriptive failure information is available for the neural network to learn. The novelty of this research is in the combined effort of fault detection and simulated data creation for improved fault recognition with the LSTM algorithm. The findings indicate that the proposed SEAD method significantly enhances classification capabilities. This

improvement is attributed to the data generated through the torque-emulating simulation model, particularly in scenarios involving increased electrical-energy demand associated with support-element failure.

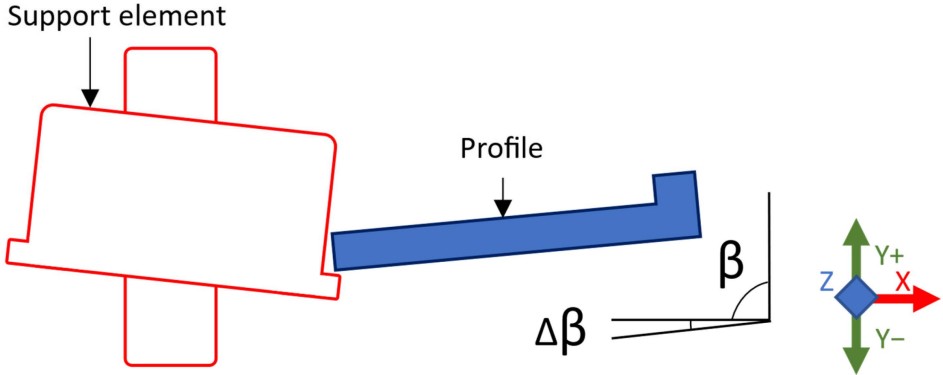

**Figure 2.** Profile angle towards the support element under support-element failure.

## 2. Simulation-Enhanced Anomaly Diagnostics (SEAD) Method

This section presents the proposed Simulation-Enhanced Anomaly Diagnostics (SEAD) method. Cases suiting the use of the explored method can be considered typical real-life applications in general, where unexpected machine anomaly symptoms are subsequently diagnosed from this process, which in this case is a machined workpiece. As described, the cause of the quality deviations in the milled material was diagnosed to be a malfunction in the support element holding the milled profile in the correct position in the milling event. The recorded timeframe contains six different milling operations, of which three original data sets (DS1…DS3) are presented herein to demonstrate the integral research outcomes.

The proposed SEAD method with physical modeling and a Long Short-Term Memory autoencoder is depicted in Figure 3. Annotations with arrows in the figure describe the information transferred between different process steps and stages (1–3) of the method.

In Stage 1, following the visual identification of the root cause of the defect, the end user estimates the timeframe for abnormal behavior in the process. This information is then employed to contextualize the condition-monitoring data from the anticipated time of occurrence. Subsequently, the simulation model is created and utilized to reproduce normal torque levels during machine operation, considering the torque calculations, which are presumably affected by the recognized fault. The simulation model is then applied to detect anomalous torque behavior within other datasets from the specified timeframe. This process facilitates the dynamic placement of anomaly thresholds for the simulated torque variable. The Prior Mean Averaged (PMA) signals are compared and based on the experimented and documented results, the machine-based data is categorized into Real Normal (RN) and Real Anomaly (RA) labels. Finally, the manual-labeling process undergoes validation in Diagnostics validation classification 1 (Class. 1) with the LSTM autoencoder to assess the model's ability to differentiate between the provided RN and RA labels.

After verifying the classification labeling achieved using the torque threshold in the simulation model, Stage 2 involves a further refinement process. This refinement includes generating Simulated Anomaly (SA) torque data using the Simulink uniform number block, in addition to Simulated Normal (SN) data. The goal is to reproduce torque signals within the normal range and to reproduce signals that surpass the predetermined anomaly threshold limit. Subsequently, the SN data and SA excitation data undergo separate validation through the LSTM autoencoder in the Simulation validation step. This assessment specifically examines the classification capability when the dataset comprises solely simulated torque data (both SN and SA), with the simulation model's input variable data acquired from the Programmable Logic Controller (PLC).

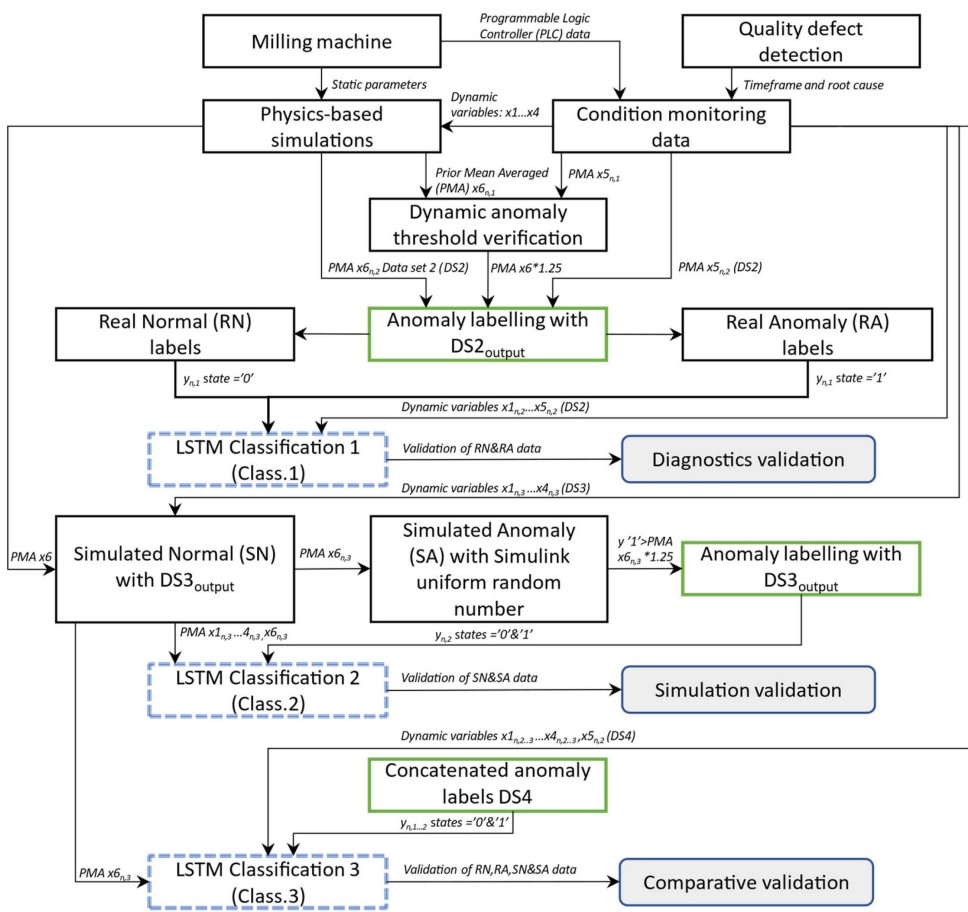

**Figure 3.** Description of the Simulation-Enhanced Anomaly Diagnostics (SEAD) method. The method operates in three distinct stages, each thoroughly explained in the subsequent chapters.

During Stage 3, the concatenation of input datasets DS2 and DS3 from previous validation rounds takes place to assess the model's performance utilizing an expanded training dataset. This phase, denoted as Comparative validation, entails the classification of the combined dataset and a subsequent analysis of its impact on the preceding validation results.

## 3. Physics-Based Simulation Modeling and Anomaly Detection

Upon detecting quality issues in the manufacturing process, data from the PLC system was collected that covered the time period when the anomaly was present. The simulation model was constructed to replicate the machine's normal torque excitation behavior to create a baseline of the machine's normal operation. Subsequently, the constructed simulation model was employed to compare the PLC data to the machine's normal torque excitation behavior.

### 3.1. Modeling of Torque Signal

The detection process was initiated by replicating the peripheral milling machine's behavior in terms of a torque signal. The spindle torque was precalculated and constructed in the Matlab-Simulink environment. The mathematical foundation for the simulation-model-torque calculations is formed in [17], where the detailed torque ($M_c$) calculations are explained. The motor-torque formula used in the simulation model construction is according to [18], as presented in Equation (1):

$$M_c = \frac{P_c \times 30 \times 10^3}{\pi \times RPM} \tag{1}$$

The net power ($P_c$) calculation is formed from the peripheral milling machine static parameters representing specification-related input parameters, such as feed material properties and physical dimensions of the spindle. The dynamic variables consist of the condition-monitoring-originated variables, including specific cutting force ($k_c$), table feed ($v_f$), axial depth of cut ($a_p$), and radial depth of cut ($a_e$). The spindle rotational speed ($RPM$) is derived from the cutting speed ($v_c$) received from the PLC system, with the relation explained in [19].

Individually recorded data sets in the condition-monitoring data contain data from five manufacturing-process variables, subsequently named variables x1...x5 in Table 2. Data set-variable annotations for the variables x1...x5 are acquired from the milling machine's PLC system, with a data-collection frequency of 5 Hz. The variables x1...x4 are: table feed, cutting speed, axial depth of cut, and radial depth of cut, respectively. The x5 variable is the measured real-torque (%) value and x6 is the simulated-torque (%) excitation, with the latter constructed based on the inputs x1...x4. Both of the torque values are described as percentual use of the electric motor's nominal torque.

**Table 2.** Data set-variable annotations.

| Variable | Description | Unit |
|----------|-------------|------|
| x1 | Table feed | (mm/min) |
| x2 | Cutting speed | (m/min) |
| x3 | Axial depth of cut | (mm) |
| x4 | Radial depth of cut | (mm) |
| x5 | Real measured torque | (%) |
| x6 | Simulated torque | (%) |

The simulation model input variables x1...x4 in data set 1 (DS1) are used in the model testing and refinement. DS1 inputs are visualized in Figure 4, which shows individual signal variation in the manufacturing process, especially in the radial depth of cut-input variable x4.

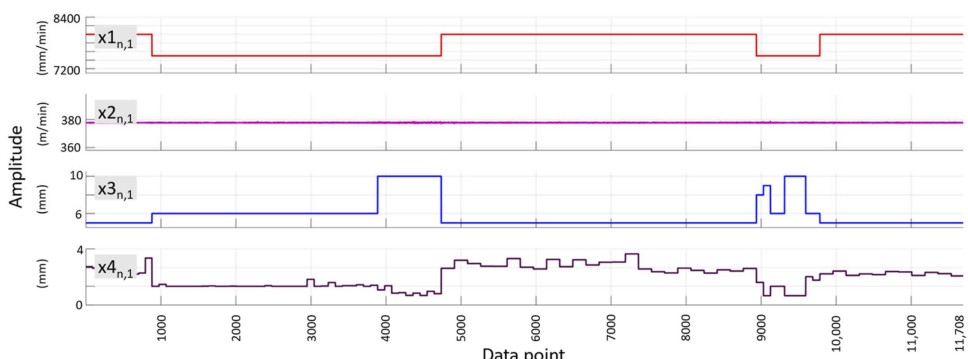

**Figure 4.** Dynamic input signals x1...x4 allocated to the simulation model in the DS1 data set.

The variables are annotated in the data set description Equation (2) in a vector format with double indexing. The first index annotates the data set-specific data point number and the second index illustrates the individual variable index, which is due to the variable usage in multiple data sets. To demonstrate, the $x1_{1,1}$ specifies the table-feed variable's first data point in the DS1 data set. The $DS1_{output}$ data consists of x5 excitation as a comparative value as well as the x6 variable excitation from the simulation-model output.

$$DS1_{output} = \begin{bmatrix} x1_{1,1} & x2_{1,1} & x3_{1,1} & x4_{1,1} & x5_{1,1} & x6_{1,1} \\ \dots & \dots & \dots & \dots & \dots & \dots \\ x1_{11708,1} & x2_{11708,1} & x3_{11708,1} & x4_{11708,1} & x5_{11708,1} & x6_{11708,1} \end{bmatrix} \quad (2)$$

The DS1$_{output}$ data set is used for verifying the simulated-signal-replication capability by comparing the simulated-torque excitation to the real signal. The temporal-simulation-model results are presented in Figure 5, where the raw signal of the real torque (x5) and simulated torque (x6) trends are presented with the tested moving-average trend of 50 for x5.

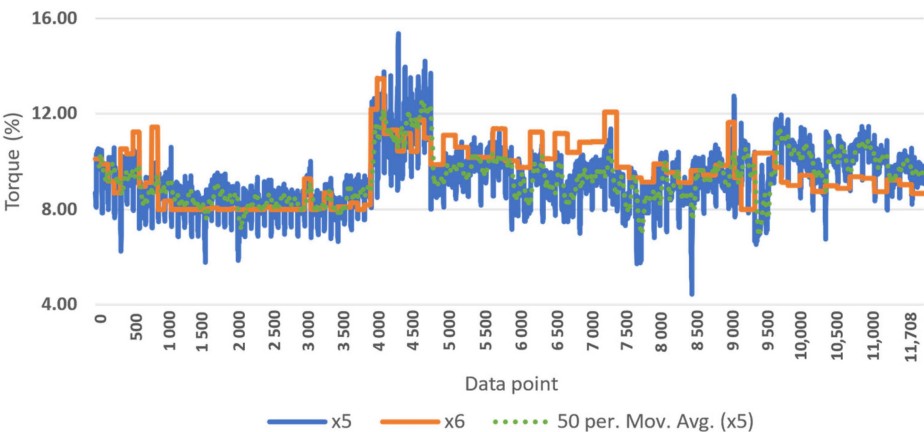

**Figure 5.** Simulated torque signal comparison to the real torque with moving average trend.

The simulation model's visual results described here give strong assurance of the model's ability to accurately replicate real signal-behavior trends with the given input variables. Simulated signals are generally created to represent real-world signals; thus, they often contain less natural variation than their real-world counterparts due to the lack of randomness or noise. Impulsive and periodical signal disturbances may appear during a machine's normal operation without relating to any real damage [18]. The descriptive statistics in Table 3 also show that the signal mean values are relatively uniform yet form slightly more varied minimum and maximum values. Despite this, the uniformity is evidenced by their mean values deviating by only 0.178, with the simulated torque (x6) mean value showing slightly higher values than the real torque (x5).

**Table 3.** Descriptive statistics of real and simulated raw torque signals.

| Signal | N | Minimum | Maximum | Mean |
|---|---|---|---|---|
| x5$_{n,1}$ | 11,708 | 4.440 | 15.350 | 9.328 |
| x6$_{n,1}$ | 11,708 | 7.980 | 13.485 | 9.506 |

### 3.2. Anomaly Threshold Classification Criteria

Despite the similarity obtained for the mean values, the boundaries for distinguishing between normal and abnormal behaviors are complex without absorbing the peak signals naturally produced by real-life applications. In the case set up with an electric motor as a torque generator, the starting torque ramp can cause load peaks when the motor starts from a standstill to obtain full-load speed. High peak currents and transient torque levels are commonly experienced during direct online starting [20]. Starting torque ramp is a type of torque control used to increase the torque of an electric motor in a linear fashion, which allows the motor to start up safely and with minimal stress on the system. Due to the aforementioned noise-increasing variables, the prior moving average (PMA) function is applied to reduce the x5 signal's low and high peak values yet minimize the effect on the mean signal amplitude. Prior moving average is a statistical term that refers to the average of a set of data from a preceding period [21], in this case, containing the average value of the previous 150 data points. The PMA procedure simultaneously adds hysteresis to the observed variable, resulting in a more consistent trend curve by reducing the size of individual peak values crossing the threshold boundary. Hysteresis is considered a control

system technique where the relationship between input and output variables involves memory effects [22]; therefore, it reduces sensitivity to noise or time lag [23]. To summarize, hysteresis helps to reduce time delays and to eliminate sharp fluctuations in system output, leading to more reliable and stable performance. Table 4 illustrates the changes to the minimum and maximum values of the trends from the PMA averaging that has only a small effect on the mean values. A small decrease in the total number of data points (N) is due to the fact that the PMA function considers the prior averaging of 150 data points, which is equivalent to 30 s of operation. Therefore, the first value in the data set is appointed at $n = 151$.

**Table 4.** Descriptive statistics of prior moving averages for real and simulated torque signals.

| Signal | N | Minimum | Maximum | Mean |
|---|---|---|---|---|
| PMA ($x5_{n,1}$,150) | 11,558 | 7.69 | 12.31 | 9.324 |
| PMA ($x6_{n,1}$,150) | 11,558 | 7.99 | 12.99 | 9.508 |

To address the natural milling-recipe changes in the simulation input variables ($x1...x4$), the threshold boundary for the anomaly distinguishing requires dynamic adjustment to the simulated signal. A dynamic threshold will allow the system to convey functional changes in the milling process and to adjust the threshold boundary accordingly without falsely detecting increased signal values as anomalistic behavior. However, determining the exact threshold limit is cumbersome due to the fact that the threshold for a dynamic torque anomaly can vary depending on the specific motor type, size, age, and environment. It has been discovered by [24] that variation in torque can be caused only by the rotor position changing by approximately 20% in normal conditions with a conventional control strategy. Therefore, combining the knowledge from previous research and from this study's experiments, the anomalistic threshold boundary is set to +25% in comparison to the $x6$ excitation received from the simulation model. The dynamic threshold is tested with the PMA-averaged data set DS1$_{output}$ as presented in Figure 6, which shows no single data point crossing the dynamic threshold.

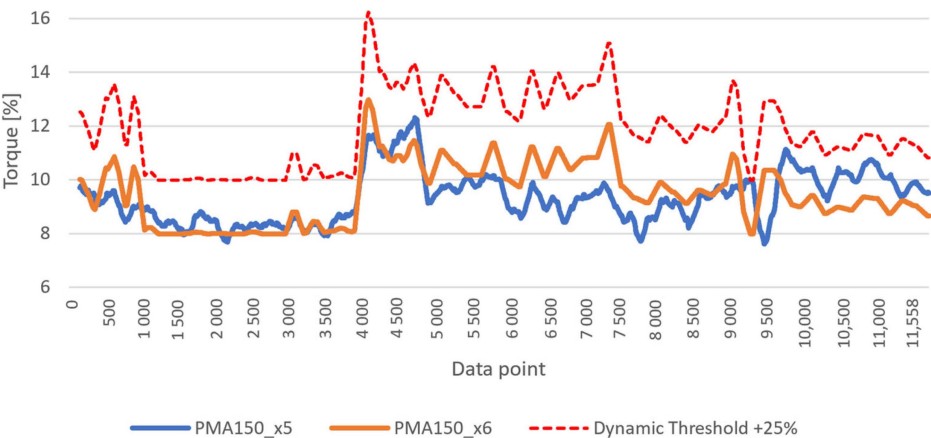

**Figure 6.** Prior moving average comparison of torque signals with dynamic anomaly-threshold boundary of +25% relative to the simulated signal x6.

### 3.3. Physics-Based Real Anomaly Detection

The simulation model is tested with the other available data sets to observe deviations from the normal machine behavior. The presented input data set DS2$_{input}$ contains

11,355 rows of data from x1...x4 that are annotated as $x1_{n,2}...x4_{n,2}$. The data set-input variables for the simulation model are presented in Equation (3) as:

$$DS2_{input} = \begin{bmatrix} x1_{1,2} & x2_{1,2} & x3_{1,2} & x4_{1,2} \\ \dots & \dots & \dots & \dots \\ x1_{11355,2} & x2_{11355,2} & x3_{11355,2} & x4_{11355,2} \end{bmatrix} \tag{3}$$

The simulation process was scrutinized for all potential data sets containing anomalous data, yet only one of the data sets showed threshold-exceeding values. The data points exceeding the dynamic anomaly threshold are visualized in Figure 7, where the PMA150-averaged real signal x5 exceeds the simulated-signal threshold within five different time intervals.

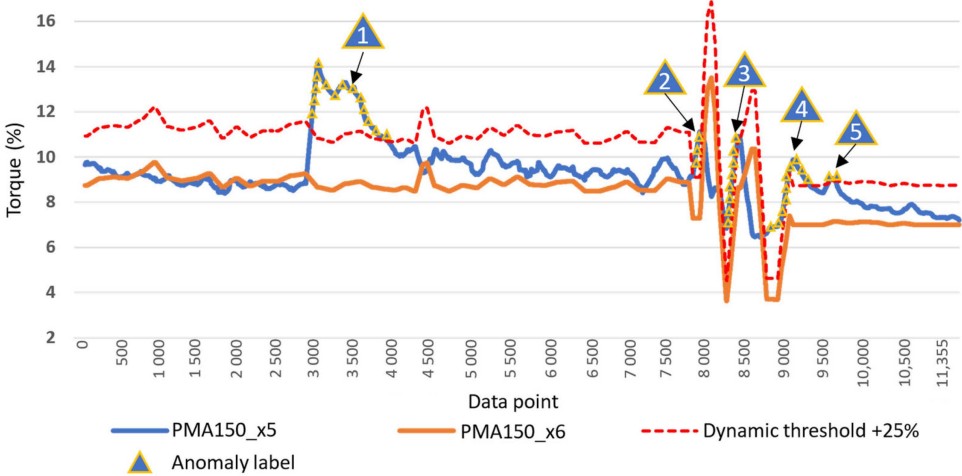

**Figure 7.** Detected anomaly label areas (numbers 1–5) according to the simulation-model threshold limits.

The anomaly occurrences in Figure 7 are labeled with yellow triangular symbols to showcase the x5 trendline points exceeding the dynamic anomaly threshold of +25% [x6n, 2 × 1.25]. Due to the DS2 data set being the only one showing quantified deviations to the simulated signal, a conclusion of credible anomaly data detection and labeling is justifiable. The details of the different real anomaly (RA) labels and the anomaly-label data set average deviation relative to the dynamic threshold limit are shown in Table 5.

**Table 5.** Details of the identified RA labels in DS2 with average deviation percentage to the dynamic threshold limit.

| Anomaly Label | Start Label | End Label | Total Anomaly Labels | Avg. Threshold Dev. (%) |
|---|---|---|---|---|
| 1 | 3049 | 4040 | 991 | 14.0 |
| 2 | 7945 | 8058 | 113 | 11.9 |
| 3 | 8320 | 8552 | 232 | 19.2 |
| 4 | 8860 | 9445 | 585 | 19.8 |
| 5 | 9678 | 9774 | 96 | 1.0 |
| | | Sum | 2017 | |

Following the physics-based anomaly detection, the total number of anomalies labeled in the DS2 data set encloses a total of 2017 RA labels (17.8% of the total data points in DS2) and 9338 RN labels (82.2%). To conclude the labeling process, the DS2 data set normal and anomaly states are summarized in Table 6.

**Table 6.** Labeled machine x6 data normal and anomaly states of the DS2.

| Total Data Points | (%) | Normal State (RN) | (%) | Anomaly State (RA) | (%) |
|---|---|---|---|---|---|
| 11,355 | 100 | 9338 | 82.2 | 2017 | 17.8 |

Hence, the output data set $DS2_{output}$ from the RA diagnostics is written in Equation (4) as follows:

$$DS2_{output} = \begin{bmatrix} y_{1,1} & x1_{1,2} & x2_{1,2} & x3_{1,2} & x4_{1,2} & x5_{1,2} & x6_{1,2} \\ \dots & \dots & \dots & \dots & \dots & \dots & \dots \\ y_{11355,1} & x1_{11355,2} & x2_{11355,2} & x3_{11355,2} & x4_{11355,2} & x5_{11355,2} & x6_{11355,2} \end{bmatrix} \quad (4)$$

The data point state column is symbolized with the letter $y$ in the data set $DS2_{output}$. The classification state of '0' is associated with 'normal' operation, whereas '1' indicates an anomaly in the operational behavior.

In summary, the DS2 remain the only data set of the original six data sets where threshold-exceeding anomalies were discovered; therefore, the validity of the diagnostics labeling results remains reliable. Thus, the simulation model has been proven to accurately detect anomalous behavior using the dynamic threshold boundaries set for anomaly detection.

## 4. Validation with the Long Short-Term Memory Autoencoder

The SEAD method uses three different validation stages to quantify the neural network's classification performance. The validation stages are:

1. Diagnostics validation;
2. Simulation validation;
3. Comparative validation.

The Diagnostics validation will assess both the anomaly detection and manual-labeling processes, examining the effectiveness of normal and anomalous labels. This evaluation will be conducted with the dynamic anomaly-threshold boundary set to analyze the x5 data. The identified anomaly labels will connect the selected PLC input data to the recognized anomalies, facilitating the diagnosis of the faulty state. Simulation validation will assess the simulated-data quality and congruency when only simulated SN and SA torque x6 states are present. In the Comparative validation, the data merging and its effect on the algorithm's classification capability will be examined.

### 4.1. Diagnostics Validation with RN and RA Data

The Diagnostics validation is performed using the x1...x5 inputs given in the $DS2_{output}$ data set, where the x5 indicates the real torque excitation of the machine while in operation. This validation processes only the $y_{n,1}$ variable and the related $x1...x5_{n,2}$ variables, excluding the x6 signal, to evaluate the labeling process of the condition-monitoring data obtained. The validation of the anomaly labeling is evaluated using an LSTM autoencoder-neural network's capability to differentiate between the labeled normal and anomalistic states. The overall algorithm structure used in this study is illustrated in Figure 8, where the variable x6 is also annotated as it is considered within the Simulation validation and the Comparative validation of the model.

The input layer consists of the x-variables annotated in Table 2 and the $y$-variable comprising the supervised label states of the machine condition. The data-split size between the training and test data in each validation is 80/20. In the encoder layers, the dropout function (DO) rate of 20% is enabled to randomly drop neurons out of the neural network. This helps to reduce model overfitting, as the network can learn from different combinations of neurons being dropped out [25]. It essentially governs the magnitude of changes applied to the model parameters. The network epochs are set to 50, with a batch size of 64 indicating the number of samples before the model weights are updated. The learning

rate used is 0.0001, which determines the extent of adjustments made to the model during each update of its weights in response to the estimated error. The latent vector of the autoencoder represents the input data in a compressed form, where only the high-level data features are retained and summarized from the input layer [26]. The decoder layers are responsible for reconstructing the original data from the latent vector. It takes the compressed representation of the input data and uses it to reconstruct the original data. This is achieved by passing the latent vector through decoder LSTM layers 1 and 2 to create a reconstructed output that is as close as possible to the original data. All the layers are accompanied by a rectified linear unit (ReLU) activation function, which helps to reduce the variable space used; therefore, increasing the training speed as the search space becomes significantly smaller in a neural network [27]. The output layer of the LSTM autoencoder represents the prediction of the encoded features, yet in this research, the machine state prediction $\hat{y}$ is observed.

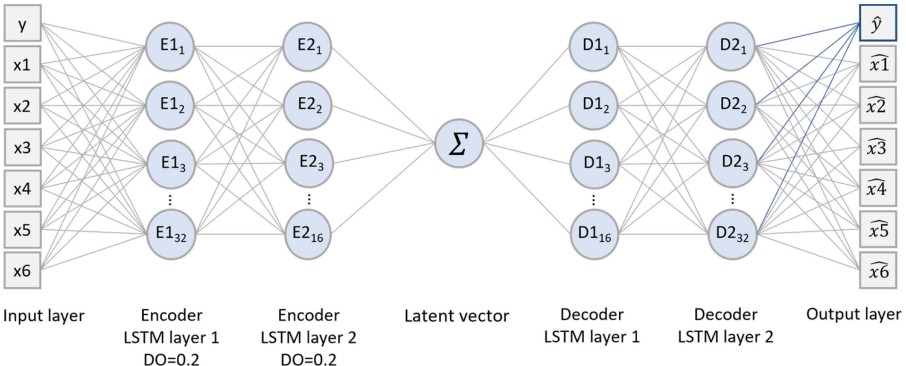

**Figure 8.** Overall structure of the supervised LSTM autoencoder used to predict and classify state $\hat{y}$.

The reconstruction error of an LSTM autoencoder measures the degree of similarity between the input and output data of the autoencoder, which is widely computed as the mean square error (MSE) of the difference between the original input and the output [28] in Equation (5). It is especially valuable for anomaly detection because the squared part of the equation can amplify errors [29].

$$K(y, \hat{y}) = || \, y - \hat{y} \, ||^2 \tag{5}$$

The reconstructed and classified data points for $y$ in DS2$_{output}$ are shown in Figure 9. The threshold boundary in the reconstruction error refers to a point at which the autoencoder is distinguishing between normal and anomalistic data [30]. The threshold boundary for the reconstruction error is typically determined through the tuning of different constants for optimal fault-detection accuracy [31], as different values may yield different levels of accuracy in the reconstruction. The empirically found threshold level of 0.3 is selected to minimize false-negative values from the results.

The preliminary Diagnostics validation results for classifying RN and RA are presented in a confusion matrix format. The confusion matrix is a useful tool for evaluating the performance of a machine learning classification algorithm by considering the importance of both positive and negative results [14]. The four outcomes are true positives (TP), false positives (FP), true negatives (TN), and false negatives (FN). The columns of the matrix represent the predicted values, and the rows represent the actual values [32], as illustrated in Table 7.

**Table 7.** Confusion matrix of the Diagnostics validation (Class. 1).

| TP | 1779 | 83 | FP |
|---|---|---|---|
| FN | 373 | 37 | TN |

The model's classification capability for the TP's is considered sufficient with this relatively low false-positive rate. However, the visual observation of the negative ratio between the true-negative and false-negative categorizations appears insufficient with the current data set where only x5-based torque values and anomaly states are present in addition to the variables x1…x4. A comparison of the classification results and a more detailed analysis of different analytical metrics are presented in the Comparative validation results.

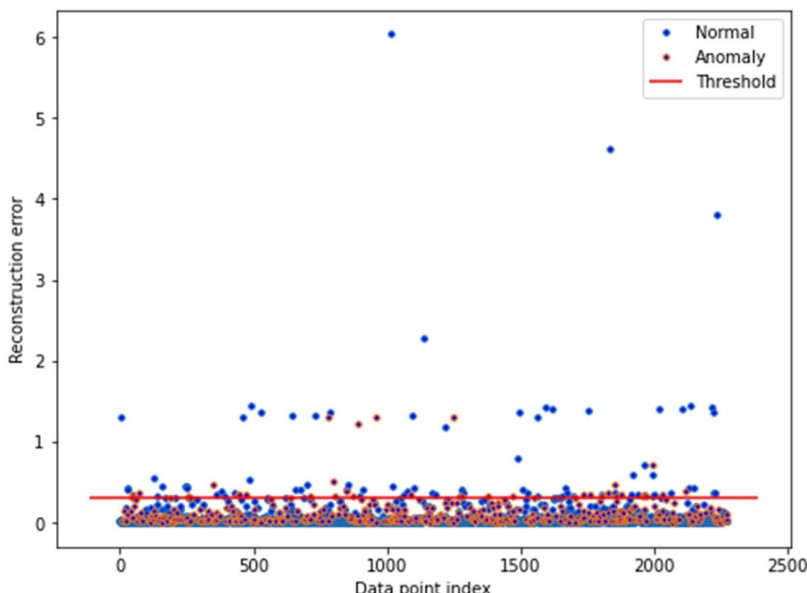

**Figure 9.** Reconstruction error with a threshold of 0.3 for the Diagnostics validation $DS2_{output}$ data set.

*4.2. Simulation Validation with SN and SA Data*

The SA data was generated using the Simulink uniform-random-number block with a multiplication range from 0.8 to 1.3. Any occurrences more than 1.25 (+25%) times the SN signal x6 were recorded and annotated as an anomaly label SA. The model-generated SA occurrences were randomly assigned to the x6 data column presented in $DS3_{output}$ Equation (6). However, the number of randomly generated x6 torque anomalies was governed by the anomaly partition depicted in the Diagnostics validation section using the DS2 data set.

$$DS3_{output} = \begin{bmatrix} y_{1,2} & x1_{1,3} & x2_{1,3} & x3_{1,3} & x4_{1,3} & x6_{1,3} \\ \dots & \dots & \dots & \dots & \dots & \dots \\ y_{11470,2} & x1_{11470,3} & x2_{11470,3} & x3_{11470,3} & x4_{11470,3} & x6_{11470,3} \end{bmatrix} \quad (6)$$

In $DS3_{output}$, the total anomaly share of 17.8% of the total data points resulted in the 2042 anomalistic data points appointed to column x6, considering the PMA-averaging function for the signal. The share of binary classified states for the $DS3_{output}$ is summarized in Table 8.

**Table 8.** Labeled simulated x6 data normal and anomaly states of the $DS3_{output}$.

| Total Data Points | (%) | Normal State (SN) | (%) | Anomaly State (SA) | (%) |
|---|---|---|---|---|---|
| 11,470 | 100.0 | 9428 | 82.2 | 2042 | 17.8 |

As was done for the Diagnostics validation, the results from the LSTM autoencoder classification for the simulation data (Classification 2) are presented in a confusion matrix format in Table 9.

**Table 9.** The simulation validation (Class. 2) confusion matrix.

| TP | 1895 | 15 | FP |
|---|---|---|---|
| FN | 302 | 81 | TN |

The results in the confusion matrix show that the simulated torque signal $x6$ in both SN and SA categories can be reliably created. The true-positive value remains dominant in the matrix yet it shows an improvement in negative classification (TN/FN) ratios. However, a direct comparison of the results between data sets $DS2_{output}$ and $DS3_{output}$ is cumbersome due to the differences in the data set-input-variable statistics. Similar to the the Diagnostics validation-results, a more comprehensive analysis of the results is performed in the Comparative validation-results.

*4.3. Comparative-Validation Results*

The Comparative validation of the data sets is performed to evaluate how the simulated data affects the model's classification capability. To assess the change, the validation is performed with the same LSTM architecture used in the diagnostics validation and Simulation validation sections, where the data sets 2 and 3 were individually tested. The term comparative validation is used to describe a process of verifying the accuracy of two or more models, processes, or systems against each other by validating the effects of the compared methods [33]. In this study, the Comparative validation stage concatenates the previously validated $DS2_{output}$ and $DS3_{output}$ data sets into a DS4 data set to evaluate the simulated data influence on the classification score. To determine the status of origin for such classification testing, the LSTM algorithm's hidden states are reset to exclude learning from the previous validation rounds, as the dimension of the hidden states is modeled to retain information from past events [34]. Table 10 summarizes the data points and their states in the DS4 data set after the data concatenation process.

**Table 10.** The number of normal and anomaly states in the DS4 data set.

| Total Data Points | (%) | Normal State (RN + SN) | (%) | Anomaly State (RA + SA) | (%) |
|---|---|---|---|---|---|
| 22,825 | 100.0 | 18,766 | 82.2 | 4059 | 17.8 |

In data set DS4 Equation (7) the variables $y$ and $x1 \ldots x4$ are positioned in congruent column positions as per the previous validation rounds; however, the collection of torque excitation datapoints is conjugated to $\{x5_{n,2}, x6_{n,3}\}$. This is done to establish a single column library for the final classification task, including excitation values from both the real and simulated signals, containing all designated states of RN, RA, SN, and SA. All the row-specific data in the DS4 data set remain synchronized, yet the order of the individual data rows is shuffled to randomize the data points used for the LSTM training and testing. The Comparative validation data set DS4 is formed as presented in Equation (7).

$$DS4 = \begin{bmatrix} y_{1,1..2} & x1_{1,2..3} & x2_{1,2..3} & x3_{1,2..3} & x4_{1,2..3} & \{x5_{1,2}, x6_{1,3}\} \\ \ldots & \ldots & \ldots & \ldots & \ldots & \ldots \\ y_{22825,1..2} & x1_{22825,2..3} & x2_{22825,2..3} & x3_{22825,2..3} & x4_{22825,2..3} & \{x5_{11355,2}, x6_{11470,3}\} \end{bmatrix} \tag{7}$$

The Comparative validation results together with more comprehensive results from the Diagnostics validation and Simulation validation are presented in Table 11. Metrics to measure the binary classification method success are generally recognized to include accuracy, precision, recall, F1-score, and area under the receiver operating characteristics curve (AUC) [13,32,35,36]. Accuracy is the number of correctly classified samples from all samples present in the test set. Precision measures the proportion of correctly identified events relative to all events identified, while recall measures the proportion of correctly identified events relative to all events that should have been identified [14]. Similar to the

other scoring metrics, the F1-score (also known as the F-measure) ranges from zero to one, but it is considered a more comprehensive measure to evaluate the model's performance due to its harmonic mean of recall and precision metrics.

**Table 11.** Validation-performance results using the Class. 1, 2, and 3 classifications containing the summation of the results.

| Validation | Accuracy | Precision (pos) | Precision (neg) | Recall (pos) | Recall (neg) | F1-Score |
|---|---|---|---|---|---|---|
| Diagnostics validation (Class. 1) | 0.799 | 0.955 | 0.090 | 0.827 | 0.308 | 0.886 |
| Simulation validation (Class. 2) | 0.862 | 0.992 | 0.211 | 0.863 | 0.844 | 0.923 |
| Comparative validation (Class. 3) | 0.845 | 0.990 | 0.170 | 0.847 | 0.789 | 0.913 |
| $\Sigma$ | $\frac{TP+TN}{TP+FP+TN+FN}$ | $\frac{TP}{TP+FP}$ | $\frac{TN}{TN+FN}$ | $\frac{TP}{TP+FN}$ | $\frac{TN}{TN+FP}$ | $\frac{2\times Precision\times Recall}{Precision+Recall}$ |

The precision and recall values are given both in positive and negative correlations due to the interest in observing both the normal operation as well as the anomaly classification capability of the algorithm. The negative-precision (neg) and recall (neg) scores in the Diagnostics validation are low; however, the relatively high overall balance presented by the F1-score suggests that the diagnostic process can successfully classify the machine states. The performance results obtained by the Simulation validation are the highest overall, which is as expected due to the lack of randomness or noise in the signal. The division between normal and anomalistic behavior was separated with the +25% dynamic threshold limit in correlation with the simulated torque x6, which works in favor of the absence of noise characteristics typical for real-life signals. In particular, the Recall (neg) score's increase between Class. 1 and Class. 2 is conspicuous and shows a significant positive difference in the model's failure-class identification capability.

The classification process for the Comparative validation was iterated multiple times due to the nature of the algorithm's function to randomly select the training and test sets. The results present an average score received from the iterations. The model's classification capability improvement is evident from the quantified measures presented in Table 11. In addition to the aforementioned scoring metrics, the AUC has become increasingly popular for measuring the performance of binary classifiers [37]. The graphical results of the LSTM classification shown in Figure 10 also include the numeric measure of the AUC. Notably, the quantified confusion matrix results are not directly comparable between the data sets Figure 10a,c due to the differences in data set sizes, yet a significant relative reduction in the false-positive class and the distinct increase in the true-negative class is observable. The receiver operating characteristics (ROC) curve visualizes the function of the true-positive rate and false-positive rate against the discrimination threshold between zero and one [38]. This discrimination threshold line is presented in 45-degree red diagonal line in Figure 10b,d. The AUC values closer to 1 emphasize better classification capability, whereas results at 0.5 indicate the model's poor capability to classify the data points [39]. By observing the AUC score, the improvement from the Diagnostics validation (AUC = 0.774) (Figure 10b) to the Comparative validation (AUC = 0.928) (Figure 10d) is generally comparable due to inclusion of DS2 data; thus, showing the model's above average increase in classification capability with the help of simulated (SN and SA) x6 torque data.

To summarize, the quantified and visualized results demonstrate improvement in the model's classification capability following the inclusion of the simulation data.

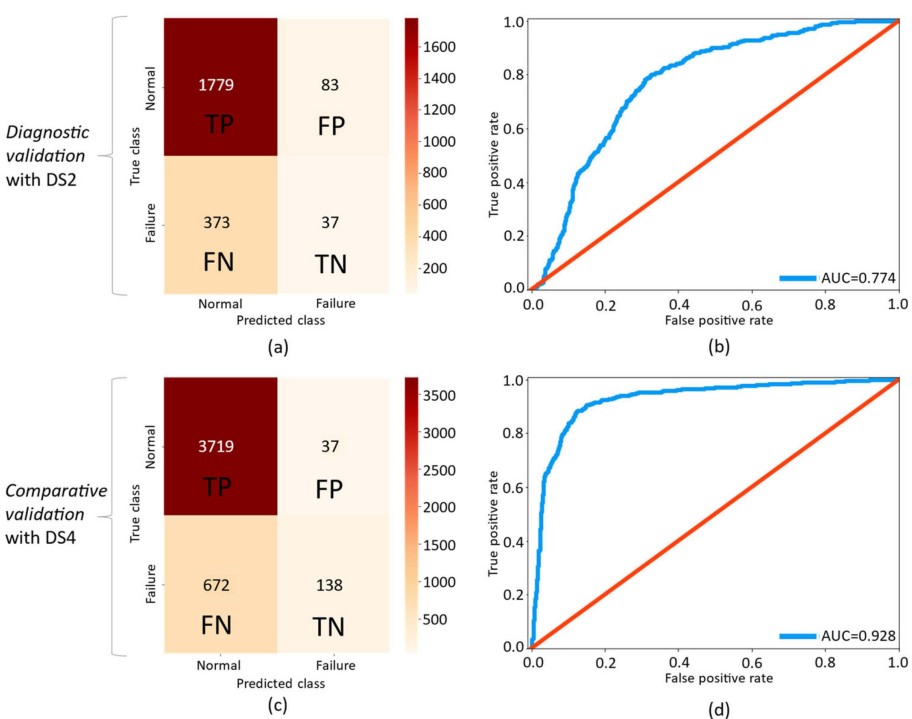

**Figure 10.** Illustration of the classification result improvement from Diagnostic validation results (**a**,**b**) to Comparative validation (**c**,**d**).

## 5. Discussion

This study proposed a SEAD method, with the benefits of utilizing the simulation model not only intended for the detection anomalies but also for the improvement of failure classification capability of a deep learning algorithm to diagnose identified deviating behavior in the manufacturing process. The simulation model was successfully utilized to detect variances in torque performance, thus connecting the machine behavior with the recognized anomalistic symptoms. Following the failure signal detection, the model-based approach was used to create a normal operational baseline to differentiate torque-origin symptoms in the machine's behavior during the support-element failure. Subsequently, the simulation model was utilized to create additional failure data for the training of the Long Short-Term Memory (LSTM) autoencoder to overcome a data scarcity problem. The Comparative validation demonstrated a notably higher level of success in the model's classification capabilities when simulated data was incorporated. The evaluation metric of the area under the curve (AUC) increased from the baseline of 0.774 to 0.928 with the inclusion of the simulated data. Thus, this study has empirically demonstrated that the employment of simulated models offers a robust and scalable approach for amplifying identified anomalistic symptom behavior, consequently augmenting the classification accuracy of the deep neural network. By following the SEAD method stages, including anomalistic behavior identification and operational data acquisition, it appears that physics-based modeling with deep neural network integration may result in more accurate and reliable anomaly detection and diagnostics. This is particularly pertinent in the peripheral milling context, where reliable system performance is paramount and unplanned production stops are not tolerated. The use of simulated models to amplify anomalistic symptoms, as demonstrated in this study, provides an excellent tool for enhancing the precision of deep neural network classification and averting costly unplanned production stops. Furthermore, the implementation of SEAD is feasible in real-life scenarios, for instance, as an edge-computing component or as an embedded element in machine automation systems.

The SEAD method presented in this study may be generalized to encompass peripheral milling machines by extending its current failure diagnostic function. In future research,

the scope of the simulation model should be extended to include other failure mechanisms by creating a more comprehensive failure library for classification algorithms to identify different types of deviating behavior. This presents an opportunity for collaborative efforts and knowledge sharing within the research community. While this study suggests the development of such a library, it is essential to consider potential initiatives for open-sourcing these resources. The creation of open-source efforts related to the milling machine failure library would not only enhance available prediction capabilities but would also foster collaboration and accelerate advancements in the field. However, to apply SEAD across various machinery or industrial processes, it is essential to meticulously analyze the distinctive characteristics and failure modes inherent in each system. The individual nature of various anomalies may include complex characteristics associated with the deviating behavior, thus limiting the physical modeling of a comprehensive failure library. Despite this, the use of the SEAD certainly has the potential to diagnose various peripheral milling machine failures more robustly and accurately, thereby facilitating earlier detection and more effective maintenance of any potential malfunctions that are generally represented by physical modeling and simulations. Overall, combining SEAD with the suggested open-source failure library would potentially lead to more efficient and cost-effective milling operations.

**Author Contributions:** Conceptualization, T.M.; methodology, T.M. and J.L.; validation, T.M. and J.L.; formal analysis, T.M.; investigation, T.M.; data curation, T.M.; writing—original draft preparation, T.M.; writing—review and editing, T.M.; visualization, T.M.; supervision, K.T.K.; resources, K.T.K.; project administration, K.T.K.; funding acquisition, K.T.K. All authors have read and agreed to the published version of the manuscript.

**Funding:** This research was funded by the Business Finland SNOBI project, grant number 545/31/2020.

**Institutional Review Board Statement:** Not applicable.

**Informed Consent Statement:** Not applicable.

**Data Availability Statement:** The data presented in this study are available on request from the corresponding author. The data are not publicly available due to confidentiality reasons.

**Conflicts of Interest:** The authors declare no conflicts of interest.

## Abbreviations

| | |
|---|---|
| AUC | Area under curve |
| ANN | Artificial neural network |
| Class. 1 | Refers to Diagnostics validation classification |
| Class. 2 | Refers to Simulation validation classification |
| Class. 3 | Refers to Comparative validation classification |
| DO | Dropout function |
| $DS1_{output}$ | Dataset 1 used for RN data diagnostics |
| $DS2_{input}$ | Dataset 2 input to the simulation model |
| $DS2_{output}$ | Dataset 2 output with state labels, including RN and RA behavior used for Diagnostics validation |
| $DS3_{output}$ | Dataset 3 output with state labels, including SN and SA behavior used for Simulation validation |
| DS4 | Dataset 4 combining $DS2_{output}$ and $DS3_{output}$ data sets used in Comparative validation |
| FN | False negative |
| FP | False positive |
| $K$ | Difference between the original input and output |
| $k_c$ | Specific cutting force (N/mm) |
| LSTM | Long short-term memory |
| $M_c$ | Torque (Nm) |
| $M_c$ [%] | Torque representation in % of the electric motor nominal torque (%) |

| | |
|---|---|
| ML | Machine learning |
| MSE | Mean squared error |
| N | Total number of data points |
| $n$ | Individual data point |
| $P_c$ | Net power (kw) |
| PLC | Programmable logic controller |
| PMA | Prior moving average |
| RA | Real anomaly labeled $M_c$ data |
| ReLu | Rectified linear unit |
| RN | Real normal labeled $M_c$ data |
| RNN | Recurrent neural network |
| RPM | Spindle rotational speed |
| ROC | Receiver operating characteristics curve |
| RPM | Rotations per minute |
| SA | Simulated anomaly labeled $M_c$ data |
| SEAD | Simulation-Enhanced Anomaly Diagnostics |
| SNOBI | Systematic development of novel business models -project |
| SN | Simulated normal labeled $M_c$ data |
| TN | True negative |
| TP | True positive |
| VFD | Variable frequency drive |
| $x1/v_f$ | Table feed (mm/min) |
| $x2/v_c$ | Cutting speed (m/min) |
| $x3/a_p$ | Axial depth of cut (mm) |
| $x4/a_e$ | Radial depth of cut (mm) |
| x5 | Real measured torque in % of the electric motor nominal value (%) |
| x6 | Simulated torque in % of the electric motor nominal value (%) |
| y | Machine state: anomalistic state label '1' or normal state '0' |
| ŷ | Predicted state of y |
| β | Nominal support angle (degrees) |
| Δβ | Deviation to the nominal support angle (degrees) |

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
