# Peer review of "Improving Deep Learning Anomaly Diagnostics with a Physics-Based Simulation Model"

_applsci, doi:10.3390/app14020800_

Round 1
Reviewer 1 Report
Comments and Suggestions for Authors
The research explores the Simulation-Enhanced Anomaly Diagnostics (SEAD) method, aiming to bolster deep learning-based anomaly detection in industrial settings. The SEAD method operates across three key stages, leveraging a physics-based simulation model to replicate machine behavior and generate high-fidelity anomaly-related data for deep learning models. It outlines each stage's process, from anomaly detection and refining simulated data to an expanded dataset for comparative validation.
The paper emphasizing the challenge of accurately distinguishing between normal and abnormal behavior due to inherent signal fluctuations. Through validation and testing, the study demonstrates the SEAD method's effectiveness in detecting anomalies, showcasing its potential to significantly enhance classification accuracy and enable early-stage anomaly recognition in industrial processes, thus addressing the scarcity of high-quality data in real-world applications.
I would suggest a few changes in the text for clarity detailed as follows:
Change "data can be obtained from a physics-based simulation model" to "data can be obtained through a physics-based simulation model."
Change "The PMA procedure will simultaneously add hysteresis to the observed variable resulting in a more consistent trend curve by reducing individual peak values crossing the threshold boundary." to "The PMA procedure simultaneously adds hysteresis to the observed variable, resulting in a more consistent trend curve by reducing individual peak values crossing the threshold boundary."
Change "Due to the DS2 data set attending the only data set showing quantified deviations to the simulated signal…" to "Due to the DS2 data set being the only one showing quantified deviations to the simulated signal…"
In the introduction section, Provide more context on the significance of the problem your research addresses. Explain why improving anomaly detection in industrial processes is crucial and how it contributes to the broader field of high energy physics.
Elaborate on the practical implications of your findings. How can the SEAD method be applied in real-world scenarios?
Author Response
Dear reviewer,
Thank you for your valuable comments. We have prepared answers to them accordingly, please find them below.
With kindest regards, Teemu Mäkiaho (first author)
Comment 1:
Change "data can be obtained from a physics-based simulation model" to "data can be obtained through a physics-based simulation model."
Answer 1: The wording has been modified as per the suggestions.
Comment 2:
Change "The PMA procedure will simultaneously add hysteresis to the observed variable resulting in a more consistent trend curve by reducing individual peak values crossing the threshold boundary." to "The PMA procedure simultaneously adds hysteresis to the observed variable, resulting in a more consistent trend curve by reducing individual peak values crossing the threshold boundary."
Answer 2: The wording has been modified as per the suggestions.
Comment 3:
Change "Due to the DS2 data set attending the only data set showing quantified deviations to the simulated signal…" to "Due to the DS2 data set being the only one showing quantified deviations to the simulated signal…"
Answer 3: The wording has been modified as per the suggestions.
Comment 4:
In the introduction section, Provide more context on the significance of the problem your research addresses. Explain why improving anomaly detection in industrial processes is crucial and how it contributes to the broader field of high energy physics.
Answer 4: Two parts of new text is provided into the ‘Introduction’ section to provide more context and significance. The added texts are:
- In manufacturing systems, identifying potential problems early on is vital for safety and cost-effectiveness. Detecting issues promptly not only ensures a secure operational environment but also helps prevent financial losses by addressing concerns before they escalate (Russo et al., 2021). This is particularly critical in high-energy processes, where machine breakages or anomalies in machinery may pose a threat to the overall process operation and to the safety of the operating personnel.
- Also, physics-based solutions may provide additional information in the absence of real data , whereas the absence of breakdown samples prompts a need for further investigation into developing effective methods to detect anomalies in production data, rather than focusing on predicting fault events (Serdio et al., 2014)
Comment 5:
Elaborate on the practical implications of your findings. How can the SEAD method be applied in real-world scenarios?
Answer 5: A short addition to Chapter 5 has been made to illustrate how the SEAD can be implemented in real-life operations: “Following, the implementation of SEAD is feasible in real-life scenarios, for instance, as an edge computing component or as an embedded element in machine automation systems.”
Reviewer 2 Report
Comments and Suggestions for Authors
- The study focuses on a specific application - detecting and replicating a failure in the support element of a peripheral milling machine. The study should discuss the potential limitations of applying the proposed method to other industrial processes or machinery. The generalizability of the SEAD method across different domains should be addressed.
- The claim that the SEAD method can be generalized to encompass peripheral milling machines may be overly optimistic. Generalizing the approach to different machinery or industrial processes requires careful consideration of the unique characteristics and failure modes of each system. The study does not provide sufficient evidence or justification for this generalization.
- While the study suggests extending the scope of the simulation model to include other failure mechanisms, it does not explore the challenges and complexities associated with modeling a comprehensive failure library. Different failure modes may have varying characteristics, and accurately capturing these in a simulation model can be a challenging task.
- There is no mention of the specific details about the simulation model or the parameters used. Requesting more information on the simulation setup, code structure, and hyperparameters would enhance the transparency and reproducibility of the study.
- Suggesting the possibility of creating a more comprehensive failure library, the study does not mention any intentions for open-sourcing such resources. Requesting information about potential open-source initiatives related to the failure library would foster collaboration and further development in the field.
Author Response
Dear reviewer,
Thank you for your valuable comments. We have prepared answers to them accordingly, please find them below.
With kindest regards, Teemu Mäkiaho (first author)
Comment 1: The study focuses on a specific application - detecting and replicating a failure in the support element of a peripheral milling machine. The study should discuss the potential limitations of applying the proposed method to other industrial processes or machinery. The generalizability of the SEAD method across different domains should be addressed.
Answer 1: The ‘5.Discussion’ -section has been modified to address comments 1,2,3 and 5. The following paragraph has been modified to incorporate model restrictions and the challenges in generalization: “The SEAD method presented in this study may be generalized to encompass peripheral milling machines by extending its current failure diagnostic function. In future research, the scope of the simulation model should be extended to include other failure mechanisms by creating a more comprehensive failure library for classification algorithms to identify different types of deviating behavior. This presents an opportunity for collaborative efforts and knowledge sharing within the research community. While the study suggests the development of such a library, it is essential to consider potential initiatives for open-sourcing these resources. The creation of open-source efforts related to the milling machine failure library would not only enhance availability prediction capabilities but would also foster collaboration and accelerate advancements in the field. However, to apply SEAD across various machinery or industrial processes, it is essential to meticulously analyze the distinctive characteristics and failure modes inherent in each system. The individual nature of various anomalies may include complex characteristics associated with the deviating behavior, thus limiting the physical modeling of a comprehensive failure library. Despite this, the use of the SEAD certainly has the potential to diagnose various peripheral milling machine failures more robustly and accurately, thereby facilitating earlier detection and more effective maintenance of any potential malfunctions that are generally represented by physical modeling and simulations. Overall, combining SEAD with the suggested open-source failure library would potentially lead to more efficient and cost-effective milling operations.”
Comment 2: The claim that the SEAD method can be generalized to encompass peripheral milling machines may be overly optimistic. Generalizing the approach to different machinery or industrial processes requires careful consideration of the unique characteristics and failure modes of each system. The study does not provide sufficient evidence or justification for this generalization.
Answer 2: See answer 1
Comment 3: While the study suggests extending the scope of the simulation model to include other failure mechanisms, it does not explore the challenges and complexities associated with modeling a comprehensive failure library. Different failure modes may have varying characteristics, and accurately capturing these in a simulation model can be a challenging task.
Answer 3: See answer 1
Comment 4: There is no mention of the specific details about the simulation model or the parameters used. Requesting more information on the simulation setup, code structure, and hyperparameters would enhance the transparency and reproducibility of the study.
Answer 4: The mathematical formula used in the simulation setup is illustrated in Equation 1 of the manuscript with the reference to an earlier study where the mathematical foundation is constructed. Prior Mean Average (PMA) function is explained under Chapter 3.2 and the Simulink uniform random number -block in Chapter 4.2. The following text is added under Chapter 4.1 to provide additional information of the LSTM hyperparameters used: “The network epochs were set to 50, with a batch size of 64 indicating the number of samples before the model weights are updated. The learning rate used was 0.0001, determining the extent of adjustments made to the model during each update of its weights in response to the estimated error.”
Comment 5: Suggesting the possibility of creating a more comprehensive failure library, the study does not mention any intentions for open-sourcing such resources. Requesting information about potential open-source initiatives related to the failure library would foster collaboration and further development in the field.
Answer 5: See answer 1
Round 2
Reviewer 2 Report
Comments and Suggestions for Authors
Acceptable